# Curvature-Adjustable Polymeric Nanolens Fabrication Using UV-Controlled Nanoimprint Lithography

**DOI:** 10.3390/mi13122183

**Published:** 2022-12-09

**Authors:** Qiang Li, Myung Gi Ji, Ashish Chapagain, In Ho Cho, Jaeyoun Kim

**Affiliations:** 1Department of Electrical & Computer Engineering, Iowa State University, Ames, IA 50011, USA; 2Department of Civil, Construction & Environmental Engineering, Iowa State University, Ames, IA 50011, USA

**Keywords:** nanolens, curvature, nanoimprinting, viscosity

## Abstract

Nanolenses are gaining importance in nanotechnology, but their challenging fabrication is thwarting their wider adoption. Of particular challenge is facile control of the lens’ curvature. In this work, we demonstrate a new nanoimprinting technique capable of realizing polymeric nanolenses in which the nanolens’ curvature is optically controlled by the ultraviolet (UV) dose at the pre-curing step. Our results reveal a regime in which the nanolens’ height changes linearly with the UV dose. Computational modeling further uncovers that the polymer undergoes highly nonlinear dynamics during the UV-controlled nanoimprinting process. Both the technique and the process model will greatly advance nanoscale science and manufacturing technology.

## 1. Introduction

Nanoscale lenses, or nanolenses, are attracting intense research interests for their crucial roles in many fields of optics, including super-resolution imaging [1], extraordinary focusing [2], and highly efficient light harvesting and emission [3]. Recently, they have even been utilized for quantum optics [4]. Accordingly, multiple techniques have been developed for their facile realization. They include chemical growth [1,5], self-organized dewetting [6], and thermal reflow [7], just to name a few.

In most existing fabrication techniques, the in-plane characteristics of the nanolens, such as the diameter, lattice pattern, and pitch, can be precisely controlled during the initial patterning steps. They, often fall short, however, in accurate control of the “out-of-plane” characteristics of the nanolens. The most notable examples are the curvature and the height which critically govern the optical functionalities of the nanolens.

So far, control of the nanolens’ curvature or height has been accomplished by techniques such as thermal reshaping [7], adjustment of the liquid’s surface tension and volume [8], or solvent exchange [9]. They, often necessitate complex additional processes, however. Moreover, in some of them, the in-plane and out-of-plane characteristics are coupled, leading the change in in-plane characteristics (lens diameter, for example) to affect out-of-plane characteristics (lens curvature, for example) or vice versa, further complicating the realization of the nanolens.

Here, we demonstrate a new technique for the curvature-adjustable realization of polymeric nanolenses. Unlike the conventional nanoimprint lithography techniques which critically rely on the initial patterning [10,11,12,13], our new technique is based on the newly reported peculiarities in photopolymer’s interaction with elastomeric nanocavities and, more importantly, their dependence on the UV dose applied to the photopolymer during the pre-curing step [14]. It is simple yet highly capable of adjusting the curvature without affecting other in-plane characteristics. Computational modeling further reveals the underlying nonlinear dynamics of the optically controlled nanoimprinting process.

## 2. Materials and Methods

### 2.1. Process Description

Figure 1 shows the process of our curvature-adjustable polymeric nanolens fabrication technique schematically. As the polymer material, we chose NOA73, a photopolymer available from Norland Inc. NOA73 was first spin-coated on a glass substrate for 15 s at 500 rpm and then 45 s at 3000 rpm. Subsequently, the NOA73 thin film was partially cured at a precisely controlled UV dose (15 mW·cm^−2^, Bluewave 200, Dymax, Torrington, CT, USA) by varying the exposure time (0, 120, 150, and 180 s for each sample, Figure 1a). In parallel, a poly(dimethylsiloxane) (PDMS) nanocup array (750 nm in pitch, 500 nm in diameter, and ~150 nm in depth *d*) was replica molded from a polycarbonate (PC) nanodome array (Figure 1b). Then, the PDMS nanocup array was placed on the partially cured NOA73 film to form intimate contact (Figure 1c), inducing the nanoscale rise of NOA73 into the nanocup cavities. The structure of NOA 73 was completely cured by applying an additional UV dose for 90 s (30 mW·cm^−2^). Eventually, the PDMS nanocup mold was peeled off from the substrate, revealing the nanolens.

As shown in the cross-sectional view snapshot (Figure 1d), the PDMS nanocups can be underfilled with NOA73. The extent of filling and, hence, the curvature of the nanolens were ultimately determined by the UV dose given to the NOA73 film during the partial pre-curing step. Note that the curvature does not affect the nanolens’ size or vice versa.

As the final step, the NOA73 nanolenses were completely cured by another round of UV exposure. Peeling off the PDMS mold revealed the completed nanolens array (Figure 1e). The nanolens array was then examined with atomic force microscopy (AFM) in the tapping mode, with the height in the center of the nanolens denoted by *h* (Figure 1d).

### 2.2. Underlying Mechanism

Figure 2 shows the underlying mechanism of the polymer nanolens formation with UV-adjustable curvature schematically. It is well-known that NOA73 under UV illumination undergoes a so-called “oxygen-inhibition effect” by which its top surface, in direct contact with oxygen, remains fluidic even though it is subject to the highest UV intensity level (Figure 2a) [15]. Owing to the oxygen inhibition effect of the NOA73’s curing process, the thickness of the fluidic layer *t*_L_ and its viscosity available for adjusting the curvature or height of the nanolens can be precisely determined at a controlled UV dose in the pre-curing step (Figure 1a). Placing the PDMS nanocup mold directly on the fluidic layer (Figure 1c) leads to very intimate contact thanks to the highly soft and deformable nature of PDMS (Figure 2b). The resulting downward vertical force *F*_V_ will drive the fluidic NOA73 into a lateral flow, followed by an upward displacement (Figure 2c). With judicious control of the viscosity through the UV dose, one can obtain nanolenses with desired curvatures. Such optical control of the nanolens’ curvature, mediated by the UV-induced viscosity modulation, has not been reported to date to the best of the authors’ knowledge.

### 2.3. Computational Modeling

To further elucidate the physical process of the UV-controlled nanoimprinting, we carried out computational simulations. Figure 3 presents the geometric and finite element mesh models of the PDMS nanocup and NOA73 film. The contact between PDMS and NOA73 was assumed to be frictionless. A uniform pressure was applied on the PDMS nanocup in all simulations and the resulting relative material behaviors and deformations in NOA73 were simulated. To apply uniform pressure to NOA73 and focus only on the imprinting process of NOA73 film, the nonlinearities of the PDMS material were not included in this simulation. Detailed considerations of nonlinear PDMS are left for future studies. All simulations were conducted on ANSYS (release 2022 R2, Canonsburg, PA, USA).

Consistent with our experimental observations [14,16], the UV pre-curing was modeled to decrease the liquidity of NOA73 substantially. Consequently, our simulation showed that pre-cured NOA73 stopped rising at a height inversely proportional to the UV dose, leaving the PDMS nanocup underfilled. This is in contrast to the uncured, liquid-like NOA73 which filled the cavity fully (Appendix A). Thus, NOA73 is modeled as a nonlinear viscoelastic material (Young’s modulus 1600 psi) via the Prony series [17,18] with its relative modulus (α) and the relation time (τ) subject to fitting.

It is noteworthy that unlike other simulation approaches to UV nanoimprint lithography (e.g., resist filling simulation) [19], this study adopts nonlinear finite element analysis (FEA) due to the relatively low-height, dome-shaped PDMS cavity filling. In simulations, the volume of NOA73 (bottom in Figure 3) is down-pressed to fill the PDMS nanocavity in a viscoelastic manner.

As shown in Figure 3, the NOA73 film carried no pre-specified zone or features that could lead to circular deformation later, whereas PDMS has a circular nanocup geometry. Therefore, the final dome-shaped deformation of the NOA73 film can be solely attributed to the “nanoimprinting.” The results will be presented in detail in Section 3.

With experimental data, the adopted nonlinear FEA requires only two key parameters (α and τ) to be calibrated, and the resulting simulation appears reasonable (Section 4) for the purpose of this study. In the future, this study’s approach can be extended to incorporate deeper and more sophisticated approaches such as molecular dynamics simulations [20,21].

## 3. Results

### 3.1. Purely UV-Controlled Nanoimprinting Case

The completed nanolenses were examined with AFM in its tapping mode. The results are shown in Figure 4 for four different levels of UV dose. A common color scale was used to emphasize the inverse proportional relationship between the UV dose and *h*, the final center height of the nanolens.

Figure 4e shows *h* over a wider range of UV dose. In particular, within the shaded region, *h* is an almost linear function of the UV dose, with the slope at approximately −68.8 nm/(J/cm^2^). At low UV doses, the PDMS nanocups were fully filled with NOA73, leading *h* eventually to its maximum possible value, i.e., the nanocup depth *d*. As the UV dose increases, *h* decreases monolithically, eventually reaching a near-flat value of ~12 nm.

The corresponding profiles of the nanolenses (Figure 4f) clearly reveal how the nanolens’ curvature evolves with the increasing UV dose. As a first-principle analysis, we fitted the nanolens’ curvature profiles to circles. The resulting radii of curvature *R* turned out to be 343, 548, 817, and 2813 nm. For *D* = 500 nm, the corresponding f-numbers, defined as
f/# ≡ f/*D* ≡ *R*/(*n* − 1)/*D*,(1)
were 1.2, 2.0, 2.9, and 10.0, respectively, with the refractive index *n* of NOA73 set to 1.56. The results are summarized in Table 1. In Appendix B, we show that even minute changes in the curvature affected the nanolens’ optical functionality significantly.

### 3.2. Computational Modeling

The results of the computational modeling, described in Section 2.3, are presented in Figure 5. During the simulations, we iteratively adjusted the material characteristics of NOA73 to obtain the highest level of agreement with the experimentally obtained data shown in Figure 4f. In particular, we focused on the three UV dose values in Table 1.

After convergence is reached, no further deformation of the UV pre-cured NOA73 was observed. Therefore, the final deformation could safely be considered as the result of the nanoimprinting by the PDMS nanocup pressing down on NOA73. It will later be shown in Section 4 that such stabilization occurred in a matter of seconds, corroborating our experimental observations presented in Appendix C.

Figure 5a–c computationally reproduced the experimentally observed inverse proportionality between the UV dose and the nanolens’ height *h*. For a more detailed comparison between the experimental and simulation results, the center-line profiles of the three nanolenses were scanned and plotted in Figure 5d in superposition with the experimental values. Except for the areas corresponding to the PDMS nanocup’s rim, the two results exhibit good agreement, further corroborating the validity of our computational model.

The discrepancies in the rim area could be ascribed to two factors. First, the AFM scan results often become dubious at corner areas due to the finite width of the AFM probe. Second, the liquid transport near the edge protrusion often involves advection in addition to the typical shear flow [22]. Often, the edge makes the liquid’s interaction with the sidewall more complex [23]. These complications may have led to the discrepancies. As shown in Appendix B, most optical nanolens effects are governed most strongly by the center portion, which lowers the level of attention to be paid to the rim area.

Table 2 summarizes the heights of the nanolenses obtained at the three UV doses, both from the experimental and the best-fitting simulation results. The discrepancy was < 5% in the two low UV dose cases. At the higher UV dose of 2.8 J/cm^2^, it increased to 16% but the overall agreement in Figure 5d still seems acceptable.

## 4. Discussion

The main parameter, or the material characteristic, that we adjusted to match the simulation results to the experimental ones in Figure 4 was the shear modulus of the partially pre-cured NOA73. To incorporate the dynamic, time-varying nature of the shear modulus into the simulation, we adopted the Prony series model in which the shear stiffness G is given as a temporal function as follows [17,18]:(2)$$\frac{G\left(t\right)}{G_{0}}=\left[\alpha_{\infty }+\sum_{i=1}^{N_{p}}\alpha_{i}e^{-\frac{t}{\tau_{i}}}\right]$$
where $\alpha_{i}$ and $\tau_{i}$ stand for the relative moduli and the relation time, respectively, for the *i*-th term. The series can have up to $N_{p}$ terms. $G_{0}$ is the initial shear stiffness. In this work, we adopted a single-term Prony series ($N_{p}=1$) for gleaning insights into the qualitative behavior of the partially pre-cured NOA73.

Some of the modeling results, which produced the best-fitting results in Figure 5, are shown in Figure 6. The vertical axis of Figure 6a–c represents the shear modulus of NOA73 normalized to the initial elastic shear modulus. From a mechanical perspective, this nanoimprinting experiment corresponds to the “creep” behavior where constant pressure (stress) leads to a gradual nonlinear increase in deformation (strain). In Figure 6, α physically implies how much shear stress is relaxed after the creep-induced deformation is over. Consequently, a higher UV dose appears to be linked to a smaller α. Furthermore, τ physically means the ratio of viscosity to stiffness in a typical spring-dashpot representation of viscoelastic material models. Thus, a higher UV dose appears to be linked to a higher τ (i.e., a higher viscosity).

## 5. Conclusions

In this paper, we reported a new nanoimprinting technique for realizing nanoscale polymer lenses. The technique allowed us to control the height and curvature of the nanolens through the UV dose given to the polymeric base material (NOA73) during the pre-curing stage without affecting other structural parameters such as the nanolens’ diameter. In our work, by adjusting the UV dose between 1 to 3 J/cm^2^, we could modulate the final height of the nanolens from ~22 to ~150 nm while maintaining the diameter fixed at 500 nm. The curvature of the nanolens was also modulated, producing f/# values ranging from 1.2 to 10. To elucidate the underlying physical mechanism behind the curvature-control, we carried out numerical simulations. The resulting model, which produced good agreement with the experimental results, suggests that the shear modulus of the partially pre-cured NOA73 undergoes a temporal change with the characteristic time in the range of seconds. The overall behavior of NOA73 predicted from the computational model agrees well with physical intuition. Future work includes refinement of the computational model with higher-order viscoelasticity, nonlinear properties of PDMS, and incorporation of air trapping and compression on the theory side and hybridization with replica molding-induced triboelectric charge [16,24,25,26] on the experimental side.

## Figures and Tables

**Figure 1 micromachines-13-02183-f001:**
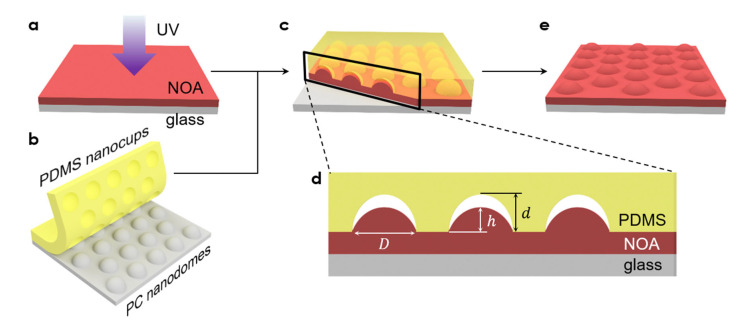
A schematic description of the curvature-adjustable polymer nanolens fabrication process. (**a**) Spin-coating of a photopolymer (NOA73) film and its partial UV pre-curing. (**b**) Replica molding of PDMS nanocups from PC nanodomes. (**c**) Forming intimate contact between the PDMS nanocup array with the partially cured NOA73 film to induce the nanoscale rise of NOA73 into the nanocup cavities. (**d**) A cross-sectional snapshot showing underfilled PDMS nanocups. (**e**) Post-curing and releasing of the PDMS mold reveal the curvature-adjusted nanolens array.

**Figure 2 micromachines-13-02183-f002:**
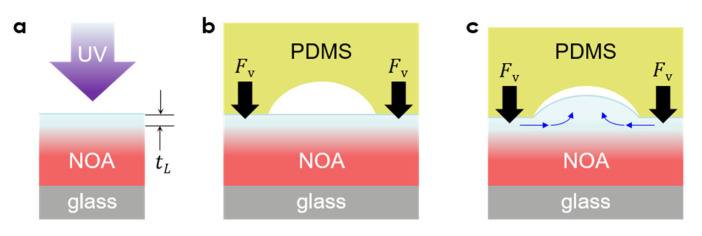
Underlying mechanism of the curvature-adjustable polymer nanolens formation. (**a**) The top surface of NOA73 remains fluidic down to *t*_L_ due to the oxygen inhibition effect. Its viscosity can be controlled through the UV dose. (**b**) Placing the PDMS mold applies downward forces *F*_V_. (**c**) A judicious control of the viscosity can ensure nanolens formation with the desired curvature.

**Figure 3 micromachines-13-02183-f003:**
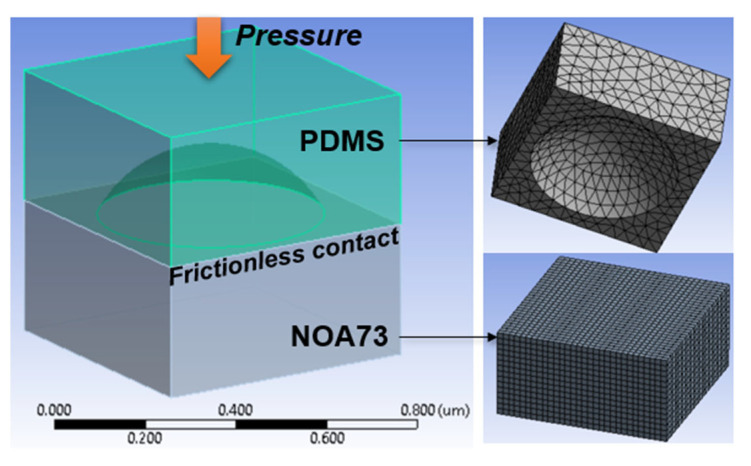
Geometry and finite element mesh models of the PDMS nanocup cavity and NOA73 film. The interface between them was modeled to be frictionless.

**Figure 4 micromachines-13-02183-f004:**
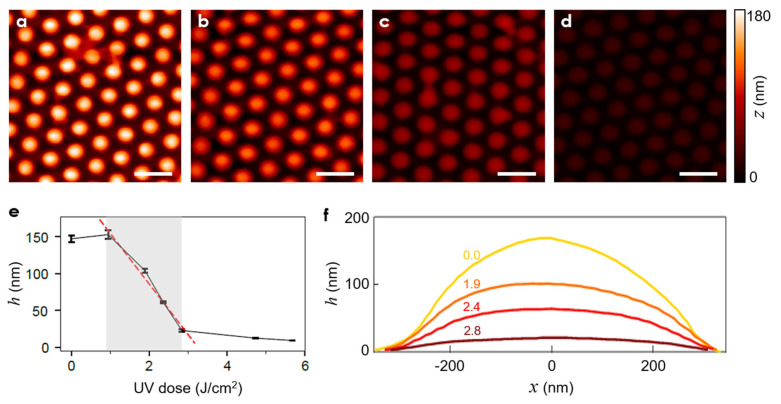
(**a**–**d**) AFM scans of the UV-controlled nanoimprinted nanolenses. The UV doses were: (**a**) 0.0, (**b**) 1.9, (**c**) 2.4, and (**d**) 2.8 J/cm^2^ (scale bars: 1.0 μm) (**e**) The final height *h* as a function of the UV dose. The bars represent the data range. (**f**) Cross-sectional profiles of the nanolenses realized with different UV dose levels (specified by the text labels).

**Figure 5 micromachines-13-02183-f005:**
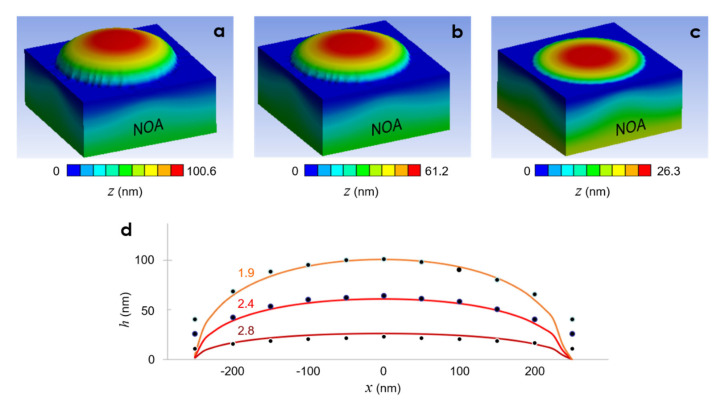
Best-fitting simulation results: (**a**–**c**) bird’s eye view plots of deformed shapes of the three different UV doses in Table 1 (1.9 J/cm^2^, 2.4 J/cm^2^, and 2.8 J/cm^2^), respectively. (**d**) Simulated cross-sectional profiles of the nanolenses corresponding to the three different UV dose levels (specified by the text labels). The dark dots represent the experimental results shown in Figure 4f.

**Figure 6 micromachines-13-02183-f006:**
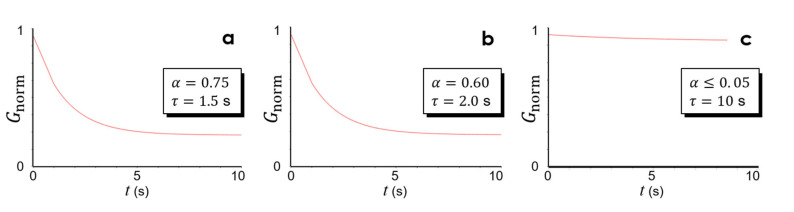
(**a**–**c**) Prony series models for the three UV doses in Table 1 (1.9 J/cm^2^, 2.4 J/cm^2^, and 2.8 J/cm^2^), respectively.

**Table 1 micromachines-13-02183-t001:** Major structural and optical characteristics of the nanolenses with different UV doses.

UV Dose (J/cm^2^)	***h*** (nm) ^1^	Radius of Curvature (nm)	f/#
0.0	147.4 ± 4.4	343	1.2
1.9	103.8 ± 2.3	548	2.0
2.4	60.6 ± 1.5	817	2.9
2.8	22.6 ± 1.4	2813	10.0

^1^ Averaged over 10 samples.

**Table 2 micromachines-13-02183-t002:** Comparison of the experimental and the best-fitting simulation results.

UV Dose (J/cm^2^)	***h*** (nm) from Experiments	***h*** (nm) from Simulations
1.9	101	100.63
2.4	64	61.21
2.8	22	26.3

## Data Availability

All experimental data are available upon request to the corresponding author.

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
