# Peer review of "Curvature-Adjustable Polymeric Nanolens Fabrication Using UV-Controlled Nanoimprint Lithography"

_micromachines, 2022, doi:10.3390/mi13122183_

Round 1

Reviewer 1 Report

You described Curvature-adjustable Polymeric Nanolens Fabrication using UV-nanoimprint process. It's many interesting paper, however, you should be described some parameters to publish your paper as follows; 

- Why did you precure of NOA73 before the imprint process using PDMS mold? 

- In simulation and experiments, there are no any recipe for the imprint process include curing time and pressure.

- Which resist filling model did you use here? In order for that, you have to consider many parameters, for example, viscosity of resist, curing time, imprinting velocity, etc. However, there are no any description for the parameters.

Author Response

We included our response to Reviewer 1 in the attached pdf file.

Reviewer 2 Report

The authors described a method of UV-controlled nanoimprint lithography for fabricating curvature-adjustable polymeric nanolens. Technically, the reported results are solid and comprehensive, which is suitable for the publication in Micromachines. The authors haven’t show sufficient supporting for clarifying the theme of the manuscript. In order to support its publication in Micromachines, the authors should also address the following points, as listed.

Recommendation: 

Publish after revisions

1. Nanoimprint is a widely used lithography method for the fabrication of micron sized structures, normally using polymers. The authors should compare the conventional lithography methods with the proposed nanoimprint method in the introduction part.

2. The authors should describe a bit more in the topic of 'nanolens' effects, which is partly mentioned in the title of the manuscript. In the discussions, the authors should display the (optical) lens-like effect of the fabricated nanolens array.

3. In Figure 5, we can find defects at the rims of the nanoimprinted nanolens. The authors should demonstrate the cause and influence of the defects.

4. Some related references should be cited properly in the reference list.

a) Holographic Resonant Laser Printing of Metasurfaces Using Plasmonic Template. ACS Photonics, 5, 5, 1665–1670, 2018.

b) Master origination by 248 nm DUV lithography for plasmonic color generation

Applied Physics Letters 118(14):141103, 2021.

DOI: 10.1063/5.0046163

Nanoimprint Lithography for Hybrid Plastic Electronics. Nano Letters 3, 4, 443–445 (2003).  In-line metrology for roll-to-roll UV assisted nanoimprint lithography using diffractometry. APL Materials 6, 058502 (2018).

Author Response

We included our response to Reviewer 2 in the attached pdf file.
